# Angiotensin-Converting Enzyme (ACE) gene polymorphism and arterial blood pressure among the Tawang Monpa of Eastern Himalayan Mountains: Is there a signature of natural selection?

Sudipta Ghosh📧 *

Department of Anthropology, North-Eastern Hill University, Shillong, Meghalaya, India

* ghosh.sudiptaa@gmail.com

## Abstract

### Objectives

The present paper aims to characterize the Angiotensin-converting enzyme (ACE) genotype, with particular emphasis on its association with arterial oxygen saturation, arterial blood pressure, hemoglobin [Hb] concentration, and ventilatory measures among the Tawang Monpa, a high-altitude native population of the Eastern Himalaya, India.

### Methods

A cross-sectional sample of 168 Monpa participants from Tawang town, Arunachal Pradesh, India, was selected who live at an altitude of $\sim$3,200 meters (m) above sea level. For each participant, height, weight, and skinfold thickness were measured, based on which body mass index (BMI, kg/m$^2$) and percentage of body fat (%BF) were calculated. Physiological measures, such as the transcutaneous arterial oxygen saturation (SaO$_2$), hemoglobin [Hb] concentration, forced vital capacity (FVC), forced expiratory volume in 1-second (FEV$_1$), and systemic arterial blood pressure were measured. First, the peripheral venous blood samples (four ml) were drawn, and then white blood cells were separated for the ACE genotyping of each participant.

### Results

Unlike high-altitude natives from Peru and Ladakh, who exhibit high frequencies of II homozygotes, the Tawang Monpa shows a significantly high frequency of ID heterozygotes (p<0.0001). In addition, no significant association was identified between ACE gene polymorphism and arterial blood pressure, oxygen saturation at rest, vital capacity, or [Hb] concentration.

**Data Availability Statement:** All relevant data are within the paper and its Supporting Information files.

**Funding:** The author(s) received no specific funding for this work.

**Competing interests:** The authors have declared that no competing interests exist.

## Discussion

The results suggest that the association of the ACE gene with resting $SaO_2$ is inconsistent across native populations living under hypobaric hypoxia. Further, ACE I/D gene polymorphism may not be under natural selection in specific native populations, including Tawang Monpa, for their adaptation to high-altitude hypoxia.

## Introduction

The angiotensin-converting enzyme (ACE) gene encodes for an enzyme found in the endothelial cells of arterial blood vessels. ACE is a vital component of the Renin-Angiotensin System (RAS), which plays an essential role in cardiovascular homeostasis. ACE helps produce angiotensin II and the degradation of Bradykinin-like vasodilator. Angiotensin II is a vasoconstrictor, which constricts arterial blood vessels and stimulates the adrenal cortex to release aldosterone that activates the RAS. Variation in ACE accounts for approximately half of the variance in ACE plasma levels [1, 2]. In particular, an insertion/deletion (I/D) polymorphism in intron 16 plays a vital role in RAS physiological pathways [3]. Interestingly, homozygotes of the deletion (D) allele exhibit significantly higher serum angiotensin-converting enzyme levels than the insertion (I) allele homozygotes [1].

In high-altitude adaptation studies, the RAS has been considered one of the most essential physiological pathways to overcome hypobaric hypoxia [4–6]. Both systemic and tissue-specific RAS up-regulation in chronic hypoxia occurs through the activation of RAS genes, including ACE [7], whose up-regulation leads to the increased production of plasma angiotensin II. Angiotensin II then stimulates cardio-respiratory functions and plays a crucial role in fluid and electrolyte homeostasis through modulation of carotid body chemoreceptor activity and its central transduction [7]. The ACE I/D gene polymorphism has been suggested as a crucial factor in altitude adaptation and acclimatization, given the effect of the -I allele on serum ACE activity [3]. This adaptive function of low serum ACE is mediated through ACE's effects on the vascular remodeling responses to hypoxia with subsequent improvements in arterial oxygen saturation ($SaO_2$). For example, the hemodynamic consequences of ACE inhibition show beneficial effects on cardiac remodeling in heart failure [8]. More specifically, chronic changes in hemodynamic forces can result in structural alterations of the vessel wall through wall diameter and thickness changes. Studies suggest that hemodynamic forces do not solely determine changes in vascular structure and can also be aggravated by inflammatory responses. Such structural changes of the medial layer of the vascular wall during hypertension are generally termed 'remodeling,' which can subsequently lead to other vascular pathologies, including hypertension and atherosclerosis [9]. It has also been observed that patients with essential hypertension are treated with ACE inhibitors because of their effect on vascular remodeling responses [9].

The role of ACE inhibitors in human adaptation to hypobaric hypoxia still needs to be understood completely. In this regard, previous studies have reported an overrepresentation of -I allele in general and II homozygotes in particular among high-altitude native populations, including Peruvian Quechua and Indian Ladakhis [6, 3, 10, 11]. The probable adaptive advantage of the II homozygotes is that the -I/I genotype maintains higher $SaO_2$ at high altitudes [10, 12]. However, studies in various ethnic groups have shown conflicting evidence of such associations between ACE I/D gene polymorphism and $SaO_2$ [12]. Hence, such association studies from other high-altitude native populations can provide a better and more conclusive

understanding of the role of ACE gene polymorphism in human adaptation to high-altitude hypoxia. Specifically, the association of ACE gene polymorphism with different physiological attributes is inconsistent across human populations (Fig 1).

Furthermore, evidence suggests that the -I allele, particularly the I/I genotype, can have detrimental effects at high altitudes. Studies conducted among Kyrgyz and Tibetan highlanders have revealed adverse health effects of both the -I and -D alleles at high altitudes. Kyrgyz highlanders with high-altitude pulmonary hypertension (HAPH) display a threefold higher frequency of the I/I genotype than normotensive controls [13]. In addition, the I/I genotype showed significantly greater mean pulmonary artery pressure (MPAP) than the I/D or D/D genotypes [13]. Among Tibetans, the ACE-D allele is significantly associated with chronic mountain sickness (CMS) [14] and arterial hypertension only among Tibetan women [2]. Altogether, these results suggest that the adaptive role of the -I allele at high altitudes might not follow a universal adaptive mechanism pathway.

Against this backdrop, the present paper aimed to answer whether the overrepresentation of the-I allele at high altitude is reproducible in other high altitude native populations and hence a universal phenomenon or limited to few native populations and thus population specific. Moreover, the association between the -I allele and $SaO_2$ at high altitudes remains to be determined. To answer these questions, the present paper examined the association of ACE I/D gene polymorphism with arterial oxygen saturation at rest, systemic arterial blood pressure, hemoglobin concentration [Hb], and ventilatory measures among the Tawang Monpa, a high-altitude Tibetan-derived population of the Eastern Himalayan Mountains of Arunachal Pradesh, India.

## Materials and methods

### Study participants

A cross-sectional sample of 168 Tawang Monpa (80 males and 88 females) inhabiting the easternmost part of the Himalayan Mountains was recruited in Tawang Town, India, at an altitude of ∼3,200 meters (m) above sea level. Study participants who self-identified as Monpa were recruited during June-July 2016. Tawang Monpa ancestry in the region is dated to ∼1,000 years before the present, following a migration from Southern Tibet [15]. Participants were healthy, non-smokers between 18 and 35 years old born and raised above 3,000 m above sea level. Study participants provided written, informed consent according to the guidelines established by the North-Eastern Hill University (NEHU) institutional review board, Meghalaya, India.

### Phenotypic measurements

For each participant, height, weight, and skinfold measurements were taken, which was followed by the calculation of body mass index (BMI, $kg/m^2$) and percentage of body fat (%BF) [16]. Transcutaneous arterial oxygen saturation ($SaO_2$) was measured at rest using a fingertip pulse oximeter (CMS50EW, Neclife, India). Hemoglobin [Hb] concentration was measured from a fingertip blood drop using a Hemocue portable hemoglobin analyzer (Hemocue®Angelholm, Sweden). Forced vital capacity (FVC) and forced expiratory volume in 1-second ($FEV_1$) were measured on each participant following standard protocols, details given elsewhere [17]. Spirometry was conducted using a portable spirometer (Care Fusion Micro I Diagnostic Spirometer, USA).

Two consecutive readings of systolic blood pressure (SBP) and diastolic blood pressure (DBP) were taken on each participant using a sphygmomanometer and a stethoscope following a 10-minute rest in the seated position. Average measurement was used in the analysis. Blood pressure measurement was classified as normal ≤120/80 mmHg; high normal 121-139/

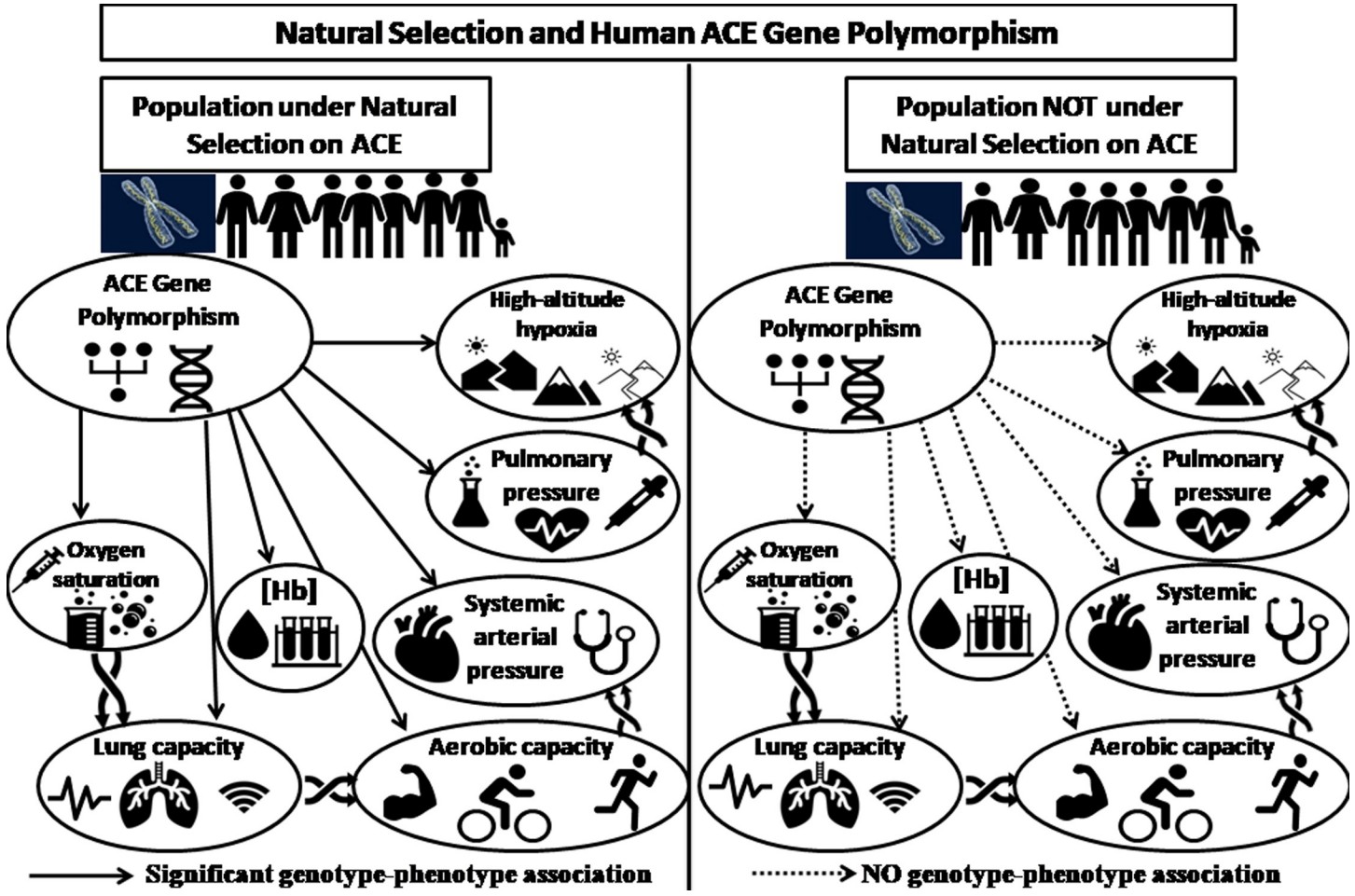

**Fig 1. ACE gene polymorphism and its association with physiological attributes in human.**

81-89 mmHg, and hypertension $\geq$140/90 mmHg, according to the American Heart Association (2005) guidelines [18]. Mean arterial pressure (MAP) was calculated using the following formula: MAP = DBP + [0.333 (SBP–DBP)].

## DNA extraction and genotyping

Peripheral venous blood samples (four ml) were drawn, and white blood cells were separated. High-molecular-weight DNA was isolated from the peripheral blood leukocytes using the Phenol-Chloroform DNA extraction method [19]. The ACE in/del allele was amplified by polymerase chain reaction (PCR) [20] using two primers, F: 5′ 3′ CTGGAGACCACTCCCATCCTT TCT and R: 5′ 3′ GATGTGGCCATCACA TTCGTCAGAT. Genotyping was conducted by 2% agarose gel electrophoresis and visualized with ethidium bromide staining.

## Statistical analysis

All statistical analyses were performed using SPSS version 24.0 (SPSS Inc., Chicago, IL, USA) and STATA 11.1 (Stata Corp LP, Texas, USA). Anthropometric and physiological variations within the population were assessed by analysis of variance (ANOVA). Intra-population variation in ACE genotype and I/D allele frequencies were calculated through chi-square

distribution. More specifically, the frequency distributions of Monpa males and females falling under the I/I, I/D, and D/D genotypes were calculated and compared statistically using the Pearson Chi-square test. The frequency distributions of–I and–D alleles were calculated and then compared with the help of the Pearson Chi-square test. The genotype-phenotype association was assessed by calculating genotype differences for all the measured variables using multivariate analysis of covariance (ANCOVA) after controlling for age and sex.

## Results

### Anthropometric, physiological, and genotypic characteristics

Statistically significant sex differences in all anthropometric and physiological variables except BMI and SaO$_2$were observed (Table 1). Although BMI was similar between males and females, % BF was significantly (p<0.001) greater in females. This result suggests that Monpa females have greater fat mass (FM) in subcutaneous and abdominal/visceral fat than males, who have higher fat-free mass (FFM). SBP, DBP, and MAP were significantly (p<0.001) higher in males than their female counterparts. Genotype data was obtained from 168 participants, which was comprised of 80 males and 88 females. Allele and genotype frequencies are shown in Table 2. The genotype frequency of I/D heterozygote (0.48) was significantly higher (Chi-square = 16.75; p<0.001) than the genotype frequencies of both I/I (0.26; p<0.001) and D/D (0.26; p<0.001) homozygotes (Table 2). The allele frequencies of the -I and -D alleles were identical (0.50; p>0.05).

### Associations between ACE gene polymorphism and physiological characteristics

No significant difference (p>0.05) among I/I, I/D, and D/D genotypes was observed for SBP, DBP, MAP, or SaO$_2$ (Table 3). D/D homozygotes have shown marginally higher (p>0.05) [Hb] concentration and FVC than both the I/I and I/D genotypes. The distributions of I/I, I/D, and D/D individuals, according to their blood pressure status (normal vs. high normal vs. hypertension), are presented in Fig 2. No statistically significant difference (p>0.05) between I/I, I/D, and D/D participants in respect of their blood pressure status was found (Table 4).

**Table 1. Descriptive statistics and comparative differences in body composition and physiological variables between the Monpa males and females.**

| Participant Characteristic | Males (N = 80) | Females (N = 88) | Total (N = 168) | P-value[b] |
|---|---|---|---|---|
| **Age (yr)** | 27.21±5.30 | 26.24±5.26 | 26.70±5.29 | 0.234 |
| **Height (cm)** | 164.45±6.29 | 153.02±5.07 | 158.46±8.05 | **<0.001** |
| **Weight (kg)** | 64.74±10.06 | 56.82±9.45 | 60.60±10.50 | **<0.001** |
| **BMI[a] (kg/m$^2$)** | 23.93±3.31 | 24.20±3.33 | 24.07±3.31 | 0.601 |
| **Body Fat (%)** | 23.42±4.73 | 35.77±3.52 | 29.89±7.44 | **<0.001** |
| **Systolic blood pressure (mmHg)** | 122.76±11.47 | 113.90±8.79 | 118.12±11.05 | **<0.001** |
| **Diastolic blood pressure (mmHg)** | 81.17±10.25 | 75.23±7.39 | 78.06±9.33 | **<0.001** |
| **Mean arterial pressure (mmHg)** | 95.03±9.87 | 88.12±7.33 | 91.41±9.28 | **<0.001** |
| **FVC[a] (l/min)** | 4.70±0.56 | 3.36±0.39 | 4.00±0.82 | **<0.001** |
| **FEV$_1$[a] (l/min)** | 3.87±0.49 | 2.81±0.39 | 3.31±0.69 | **<0.001** |
| **O$_2$ Saturation (%)** | 96.98±2.05 | 96.88±2.75 | 96.92±2.43 | 0.791 |
| **Hb[a] (g/dl)** | 15.34±1.05 | 12.40±1.49 | 13.80±1.96 | **<0.001** |

Data are means ±SD.

[a]BMI–Body mass index; FVC–Forced vital capacity; FEV$_1$ –Forced expiratory volume in 1-second; Hb–Hemoglobin concentration.

[b]Significant P-values are highlighted in bold.

**Table 2. Genotype and allele frequencies of ACE I/D gene polymorphism among Monpa, by sex.**

| Participant | ACE Genotypes[a] | | | I/D Alleles | |
|---|---|---|---|---|---|
| | I/I | I/D | D/D | I | D |
| Males (N = 80) | 0.23(18) | 0.50 (40) | 0.27(22) | 0.48 (76) | 0.52 (84) |
| Females (N = 88) | 0.28 (25) | 0.46(41) | 0.26(22) | 0.51 (91) | 0.49 (85) |
| Total (N = 168) | 0.26(43) | 0.48 (81) | 0.26 (44) | 0.50 (167) | 0.50 (169) |

Data are frequencies with counts in parentheses.

[a]Intra-population differences in ACE genotypes (I/I vs. I/D vs. D/D): $p < 0.001$

Even though the proportion of I and D alleles according to their blood pressure status indicates higher frequencies of the D allele in the 'hypertension' category (Table 4), the differences are not statistically significant ($p > 0.05$; Fig 2).

## Discussion

The evolutionary significance of sex differences in anthropometric and physiological measures is still unclear, as the trend is not comparable across human populations. However, in line with the present study, sex differences in physiological attributes were also reported in previous studies [17, 21–23].

The gene for human ACE is an autosomal gene located on chromosome 17. Thus, as expected, no sex difference was observed in ACE gene polymorphism in Tawang Monpa, as reported in other high-altitude populations [10]. Such results are significant from an evolutionary viewpoint to understand the role of gender equality in human genetic adaptation to hypobaric hypoxia. Several studies have highlighted the crucial role of ACEI/D gene polymorphism in high-altitude adaptation. The association of the -I allele with oxygen saturation in high-altitude adaptation and its overrepresentation among the high-altitude native populations have been well documented. The present study has revealed that both -I and -D alleles have equivalent frequencies among the Tawang Monpa, in line with previous findings among Tibetan [2] and Kyrgyz highlanders [24] but contrary to the reporting on other high-altitude populations like Sherpa [25], Ladakhi [6], and Peruvian Quechua[10, 11].

Initially, this equivalent representation of both -I and -D alleles in Monpa might suggest an evolutionary significance of heterozygous advantage, as found in other native highlanders. For example, I/D heterozygotes may possess an advantage over I/I and D/D homozygotes to

**Table 3. Estimated marginal means (Mean±SE) of physiological characteristics of adult Monpa by their ACE Gene Polymorphisms, while controlling for age and sex.**

| Physiological Characteristics | ACE Genotypes | | | P-value |
|---|---|---|---|---|
| | I/I (N = 43) | I/D (N = 81) | D/D (N = 44) | |
| Systolic blood pressure (mmHg) | 118.01±1.50 | 118.10 ±1.09 | 118.26 ±1.48 | 0.993 |
| Diastolic blood pressure (mmHg) | 78.56±1.29 | 77.41±0.93 | 78.76±1.27 | 0.627 |
| Mean arterial pressure (mmHg) | 91.71±1.25 | 90.97±0.91 | 91.93±1.23 | 0.794 |
| FVC[a] (l/min) | 3.96±0.07 | 4.01±0.05 | 4.02±0.07 | 0.809 |
| FEV$_1$[a] (l/min) | 3.18±0.06 | 3.36±0.05 | 3.35±0.06 | 0.057 |
| Oxygen saturation (%) | 96.91 ±0.38 | 97.01±0.27 | 96.77±0.37 | 0.872 |
| Hb[a](g/dl) | 13.77±0.20 | 13.67±0.14 | 14.06 ±0.19 | 0.255 |

P-values are at 5% probability level.

[a]FVC–Forced vital capacity; FEV$_1$ –Forced expiratory volume in 1-second; Hb–Hemoglobin concentration.

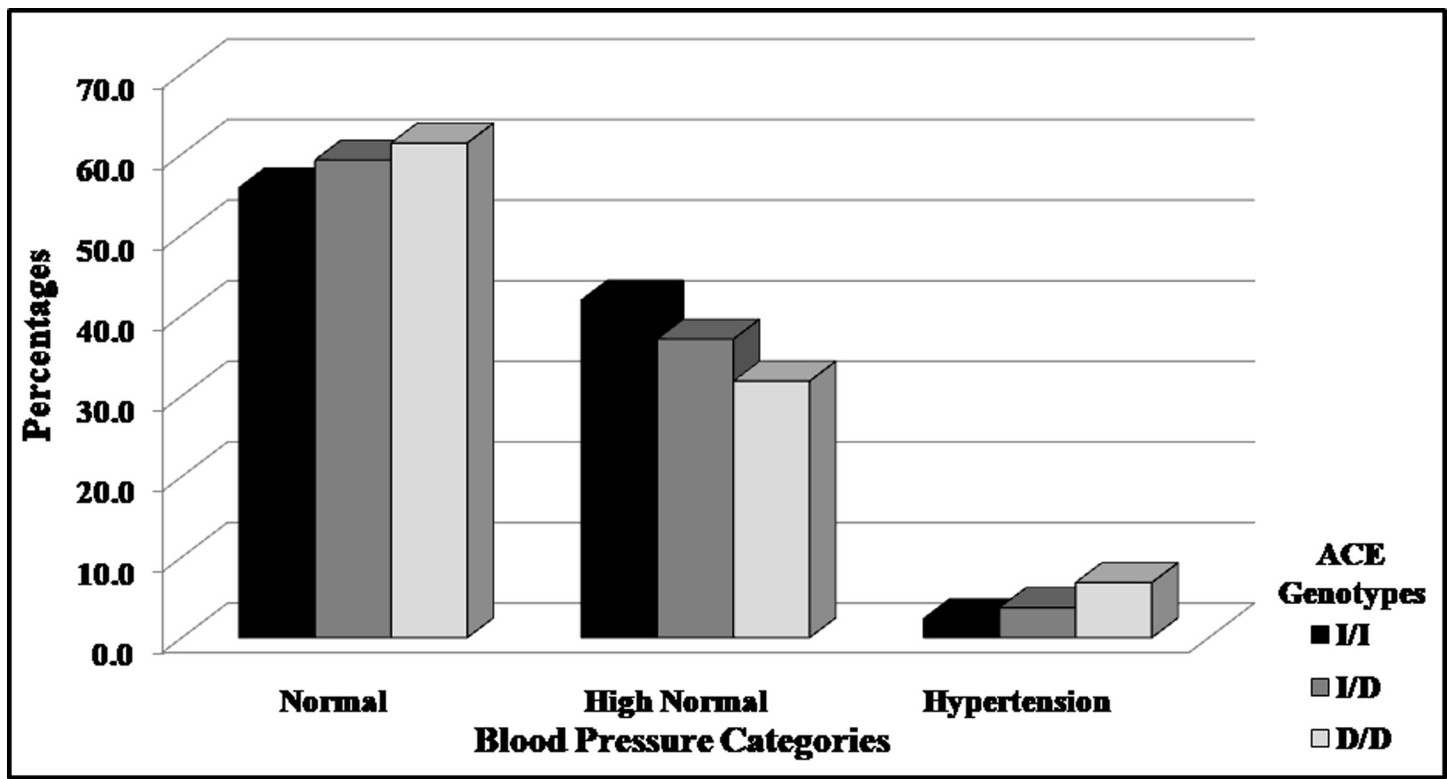

ACE genotype differences for blood pressure categories (I/I vs. I/D vs. D/D): Chi-square value= 1.845; df= 4; p value= 0.764.

**Fig 2. ACE I/D genotypes distribution according to the blood pressure categories among the Tawang Monpa.**

protect them from a combination of HAPH, CMS, and arterial hypertension. Previous studies have shown significantly greater prevalence of HAPH and associated right ventricular hypertrophy with the I/I genotype among several high-altitude native populations [5, 13, 24] and essential hypertension in Caucasians [26]. Similarly, the D/D genotype was found to be significantly associated with left ventricular hypertrophy [27–29] and essential hypertension in Asian Indians [30], African-Americans [31] as well as Caucasians [32]. Thus, the ACE gene clearly shows a heterozygous (I/D) advantage over the homozygotes (I/I and D/D) in high and low-altitude populations. Moreover, it was reported that highlander Tibetans, who have shown an overrepresentation of ACE heterozygous (I/D), have pulmonary arterial pressures similar to sea-level values. This phenomenon is considered adaptive because elevated pulmonary arterial pressure at high altitudes is a maladaptive response to chronic hypoxia [33]. Although studies have also highlighted that I/I subjects showed 23% greater aerobic fitness than their D/D counterparts at high altitudes [cf.3], the evidence is conflicting across human populations.

The Tawang Monpa did not reveal any association between resting $SaO_2$ and ACE genotype (p>0.05). A previous study among Peruvian Quechua has reported that the ACE I/I genotype is significantly associated with higher resting and sub-maximal exercise $SaO_2$ [10]. Further, the performance advantage of I/I homozygote individuals at high altitudes might be mediated through their subsequently higher $SaO_2$ [3, 12]. No significant associations (p>0.05) between ACE gene polymorphism and other physiological characteristics were observed among Monpa, which suggests that the ACE gene does not influence lung capacities and [Hb] concentration in the studied population. Hence, the lack of association between ACE genotype and

**Table 4. Distribution of blood pressure levels in adult Monpa, by ACE genotypes and alleles.**

| | | Blood Pressure Categories | | |
| | | Normal | High Normal | Hypertension |
|---|---|---|---|---|
| **ACE Genotypes** | **I/I** | 24 (55.81) | 18 (41.86) | 1 (2.33) |
| | **I/D** | 48 (59.26) | 30 (37.04) | 3 (3.70) |
| | **D/D** | 27 (61.36) | 14 (31.82) | 3 (6.82) |
| | **Total** | 99 (58.93) | 62 (36.90) | 7 (4.17) |
| **ACE Alleles** | **I** | 56 (66.67) | 24 (28.57) | 4 (4.76) |
| | **D** | 53 (63.10) | 19 (22.62) | 12 (14.29) |
| | **Total** | 109 (64.88) | 43 (25.60) | 16 (9.52) |

Data are frequencies with percentages in parentheses.

physiological attributes, including $SaO_2$ among the Tawang Monpa, points towards the fact that the polymorphisms in the ACE gene are not under the influence of natural selection in this population. More research involving a similar study design must be required before concluding anything substantially.

Literature on the association between ACE genotype and arterial blood pressure has shown contradicting evidence. For example, previous studies have reported significant associations between ACE D/D homozygotes and essential systemic hypertension in Asian Indians residing at low altitudes [30] and with HAPE among Indians at high altitudes [34]. However, in the present study, there were no significant differences in right arterial pressure between Monpa with I/I, I/D, or D/D genotype. This phenomenon indicates that the ACE gene does not play any role in maintaining the arterial blood pressure level in the studied population. Of course, the study's limitation is that pulmonary arterial pressure was not included. Similar results revealing no association between ACE I/D genotype and essential hypertension were also reported on Tibetan highlanders [2, 35], USA Caucasians [36], Japanese lowlanders [37], Dutch population [38], and other low-altitude populations [39], which probably suggests that mutations at the ACE locus do not commonly contribute to the pathogenesis of hypertension in the populations mentioned above [39]. In other words, this result further confirms the fact that ACE gene polymorphism is not under selection among the populations mentioned above, including Monpa.

Further, environmental exposure might have played a role here, as several studies mentioned above were performed under normoxic conditions. Nevertheless, the present study is consistent with other studies undertaken on high-altitude Tibetans. Limitations of the present study are that pulmonary arterial pressure and sub-maximal or maximal $SaO_2$ were not measured, which would have given more detailed information on ACE genotype-phenotype associations concerning human adaptation to high-altitude hypoxia. In addition, the sample size of the present study is on the lower side of robust analysis. Therefore, it is essential to conduct further studies with a larger sample size to understand the scenario better.

In conclusion, ACE gene polymorphism allele frequencies vary across ethnic groups (Tibetans and Tawang Monpa vs. Peruvian Quechua and Ladakhi) living under hypobaric hypoxia. There is no signature of natural selection on ACE I/D gene polymorphism among the Tawang Monpa, considering no evidence of any association of resting $SaO_2$, arterial blood pressure, [Hb], and lung functions with ACE I/D gene polymorphism. Further study is necessary to understand the effect of ACE gene polymorphism on arterial blood pressure, $SaO_2$, and [Hb] concentration in different ethnic/indigenous populations living under hypobaric hypoxia to

gain a better understanding of the role of ACE gene polymorphism on the human adaptation to high-altitude hypoxia.

## Supporting information

**S1 Data.**
(XLSX)

## Acknowledgments

I extend my gratitude to Dr. Murali Kotal, Anthropological Survey of India North-Eastern Regional Center, Shillong, for his help and support in providing me with the laboratory space for DNA extraction. I thank Prof. D.K. Limbu, Department of Anthropology, NEHU, Shillong, for his valuable editorial comments. My special thanks to all the Monpa participants for their unconditional help and support during the data collection.

## Author Contributions

**Conceptualization:** Sudipta Ghosh.

**Data curation:** Sudipta Ghosh.

**Formal analysis:** Sudipta Ghosh.

**Funding acquisition:** Sudipta Ghosh.

**Investigation:** Sudipta Ghosh.

**Methodology:** Sudipta Ghosh.

**Project administration:** Sudipta Ghosh.

**Resources:** Sudipta Ghosh.

**Software:** Sudipta Ghosh.

**Supervision:** Sudipta Ghosh.

**Validation:** Sudipta Ghosh.

**Visualization:** Sudipta Ghosh.

**Writing – original draft:** Sudipta Ghosh.

**Writing – review & editing:** Sudipta Ghosh.

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
