## [Decision Letter · Decision Letter 0]

31 Jul 2023

PONE-D-23-15795Angiotensin Converting Enzyme (ACE) gene polymorphism and arterial blood pressure among the Tawang Monpa of Eastern Himalayan Mountains: Is there a signature of natural selection?PLOS ONE

Dear Dr. Ghosh,

Thank you for submitting your manuscript to PLOS ONE. After careful consideration, we feel that it has merit but does not fully meet PLOS ONE’s publication criteria as it currently stands. Therefore, we invite you to submit a revised version of the manuscript that addresses the points raised during the review process.

ACADEMIC EDITOR: <ul><li>I hope this message finds you well.

I am writing to you regarding the manuscript titled "Angiotensin Converting Enzyme (ACE) gene polymorphism and arterial blood pressure among the Tawang Monpa of Eastern Himalayan Mountains: Is there a signature of natural selection?" submitted to PLOS ONE.

After careful review by two independent experts, and taking into consideration both the Results and Discussion sections you have provided, I would like to invite you to revise the manuscript. The reviewers, along with myself, found your work to be promising, yet there are specific areas that need to be addressed.

**Methodology and Statistical Analysis**: Your manuscript contains an extensive statistical analysis of anthropometric, physiological, and genotypic characteristics. While the analysis is robust, the reviewers suggest that it could be helpful to provide a clearer explanation of the methodology used, particularly in relation to the genotype frequency comparison (χ2=16.75; p<0.001).**Interpretation of Results**: The findings related to sex differences in anthropometric and physiological variables, as well as the ACE gene polymorphism, are interesting. However, the Discussion could benefit from a deeper analysis of how these results relate to the broader context of genetics and adaptation, especially in the Tawang Monpa population.**Figure and Table Presentation**: Please consider improving the presentation of tables and figures, ensuring that they are self-explanatory and align with the text.**Blood Pressure and ACE Genotype Analysis**: The result that there is no significant difference (p>0.05) among the I/I, I/D, and D/D genotypes in blood pressure levels and other physiological characteristics needs a more comprehensive discussion. It would be insightful to further explore and discuss why these findings are significant and how they contribute to our understanding of ACE gene polymorphism in the population studied.**Language and Formatting**: Finally, there are minor language and formatting issues that need to be addressed to enhance the readability and professionalism of the manuscript.

Enclosed, please find the detailed comments from both reviewers for your consideration. I kindly request that you revise the manuscript in line with these comments and resubmit it to PLOS ONE. Please ensure that you provide a detailed response to each reviewer's comments when you resubmit your manuscript.

Your work on this subject is indeed valuable, and I believe that addressing these comments will greatly enhance the contribution of your study to our readership.

Thank you for considering PLOS ONE for your work, and I look forward to receiving your revised manuscript.

Best regards==============================

We look forward to receiving your revised manuscript.

Kind regards,

Esteban Ortiz-Prado

Academic Editor

PLOS ONE

Additional Editor Comments:

Dear Authors, I hope this message finds you well.

I am writing to you regarding the manuscript titled "Angiotensin Converting Enzyme (ACE) gene polymorphism and arterial blood pressure among the Tawang Monpa of Eastern Himalayan Mountains: Is there a signature of natural selection?" submitted to PLOS ONE.

After careful review by two independent experts, and taking into consideration both the Results and Discussion sections you have provided, I would like to invite you to revise the manuscript. The reviewers, along with myself, found your work to be promising, yet there are specific areas that need to be addressed.

Methodology and Statistical Analysis: Your manuscript contains an extensive statistical analysis of anthropometric, physiological, and genotypic characteristics. While the analysis is robust, the reviewers suggest that it could be helpful to provide a clearer explanation of the methodology used, particularly in relation to the genotype frequency comparison (χ2=16.75; p<0.001).

Interpretation of Results: The findings related to sex differences in anthropometric and physiological variables, as well as the ACE gene polymorphism, are interesting. However, the Discussion could benefit from a deeper analysis of how these results relate to the broader context of genetics and adaptation, especially in the Tawang Monpa population.

Figure and Table Presentation: Please consider improving the presentation of tables and figures, ensuring that they are self-explanatory and align with the text.

Blood Pressure and ACE Genotype Analysis: The result that there is no significant difference (p>0.05) among the I/I, I/D, and D/D genotypes in blood pressure levels and other physiological characteristics needs a more comprehensive discussion. It would be insightful to further explore and discuss why these findings are significant and how they contribute to our understanding of ACE gene polymorphism in the population studied.

Language and Formatting: Finally, there are minor language and formatting issues that need to be addressed to enhance the readability and professionalism of the manuscript.

Enclosed, please find the detailed comments from both reviewers for your consideration. I kindly request that you revise the manuscript in line with these comments and resubmit it to PLOS ONE. Please ensure that you provide a detailed response to each reviewer's comments when you resubmit your manuscript.

Your work on this subject is indeed valuable, and I believe that addressing these comments will greatly enhance the contribution of your study to our readership.

Thank you for considering PLOS ONE for your work, and I look forward to receiving your revised manuscript.

Best regards

Reviewers' comments:

Reviewer's Responses to Questions

**Comments to the Author**

1. Is the manuscript technically sound, and do the data support the conclusions?

Reviewer #1: Partly

Reviewer #2: Yes

2. Has the statistical analysis been performed appropriately and rigorously? 

Reviewer #1: Yes

Reviewer #2: Yes

3. Have the authors made all data underlying the findings in their manuscript fully available?

Reviewer #1: Yes

Reviewer #2: No

4. Is the manuscript presented in an intelligible fashion and written in standard English?

Reviewer #1: Yes

Reviewer #2: Yes

5. Review Comments to the Author

Reviewer #1: This study delineates the distribution of angiotensin-converting enzyme (ACE) genotypes (I/I, I/D, and D/D) among 168 individuals from Tawang Monpa, a high-altitude native population from the Eastern Himalaya. It also identifies potential correlations between ACE genotypes and several physiological attributes such as oxygen saturation, blood pressure, and hemoglobin concentration. Although the topic is compelling and has previously yielded inconsistent results across different studies, the findings from this investigation are somewhat limited and do not provide a robust association between genotypes, clinical data, and high-altitude adaptation. To this end, I strongly recommend the following:

1) The number of individuals analyzed in this study is relatively small, which limits its statistical power. I would suggest either increasing the sample size or including a thorough discussion about the limitations posed by the current sample size in the discussion section of the paper.

2) Adaptation to high altitudes is a multifactorial process and is not solely linked to the behavior of the ACE genotypes. Hence, to address this substantial biological question more comprehensively, I recommend integrating an analysis of other genes that play a significant role in high-altitude adaptation, such as EPAS1, EGLN1, PDE4D, HBB, HBA1, HBA2, VEGFA, and/or PPARGC1A.

Reviewer #2: Good job.

Place the p<0.001 value symbol at each value that obtained it, in order to be consistent with the text "The genotype frequency of I/D heterozygote (0.48) was significantly higher (χ2=16.75) p<0.001) than the genotype frequencies of both I/I (0.26) and D/D (0.26) homozygotes (Table 2). The allele frequencies of the -I and -D allele were identical (0.50)."

6. PLOS authors have the option to publish the peer review history of their article (what does this mean?). If published, this will include your full peer review and any attached files.

Reviewer #1: **Yes: **Andrés López-Cortés

Reviewer #2: No

---

## [Author Response · Author response to Decision Letter 0]

2 Sep 2023

Reviewer # 1 Comments:

1) The number of individuals analysed in this study is relatively small, which limits its statistical power. I would suggest either increasing the sample size or including a thorough discussion about the limitations posed by the current sample size in the discussion section of the paper. 

Author’s response: I agree with the reviewer. The sample size of the present study is not on the upper side of the limit. However, since it is not possible to increase the sample size at this point, I have given a thorough discussion about the sample size limitation in the discussion section of the manuscript, as follows:

“......the sample size of the present study being on the lower side of robust analysis, it is important to conduct further study with larger sample size for better understanding of the scenario.”(p. 13).

2) Adaptation to high altitudes is a multifactorial process and is not solely linked to the behaviour of the ACE genotypes. Hence to address this substantial biological question more comprehensively, I recommend integrating an analysis of other genes that play a significant role in high-altitude adaptation, such as EPAS1, EGLN1, PDE4D, HBB, HBA1, HBA2, VRGFA, and/or PPARGC1A

Author’s response: I thank the reviewer for pointing out this. While agreeing with the reviewer, I want to mention here that the focal theme of the paper is to understand the role of the ACE genotype in high-altitude adaptation among humans. 

However, following the reviewer's suggestions, I will analyze the role of other important candidate genes and try to understand their roles among the Monpa in my following paper. 

Reviewer #2 Comments:

1. Place the p<0.001 value symbol at each value that obtained it, in order to be consistent with the text “The genotype frequency of I/D heterozygote (0.48) was significantly higher (χ2=16.75; p<0.001) than the genotype frequencies of both I/I (0.26) and D/D (0.26) homozygotes (Table 2). The allele frequencies of the -I and -D allele were identical (0.50).”

Author’s response: I am thankful to the reviewer for this suggestion. Following this suggestion, I have inserted “p<0.001” in the appropriate places in the above sentence.

Academic Editor’s Comments:

1. Methodology and Statistical Analysis: Your manuscript contains an extensive statistical analysis of anthropometric, physiological, and genotypic characteristics. While the analysis is robust, the reviewers suggest that it could be helpful to provide a clearer explanation of the methodology used, particularly in relation to the genotype frequency comparison (χ2=16.75; p<0.001).

 Author’s response: I have included this information in the methodology section as follows:

“.....frequency distributions of Monpa males and females falling under the I/I, I/D, and D/D genotypes were calculated and then compared statistically by using Pearson Chi-square test. Additionally, frequency distributions of –I and –D alleles were calculated and then compared with the help of Pearson Chi-square test.....”(p 7).

2. Interpretation of Results: The findings related to sex differences in anthropometric and physiological variables, as well as the ACE gene polymorphism, are interesting. However, the Discussion could benefit from a deeper analysis of how these results relate to the broader context of genetics and adaptation, especially in the Tawang Monpa population.

Author’s response: I am thankful to the academic editor for this important suggestion. Following this suggestion, I have included a brief description of this information in the Discussion section as follows:

“………..Evolutionary significance of sex differences in anthropometric and physiological measures is still unclear, as the trend is not comparable across human populations. However, in line with the present study, sex differences in physiological attributes were also reported in previous studies……..”(p 11).

3. Figure and Table Presentation: Please consider improving the presentation of tables and figures, ensuring that they are self-explanatory and align with the text.

Author’s response: I have considered this suggestion and modified the tables as well as both the figures accordingly.

4. Blood Pressure and ACE Genotype Analysis: The result that there is no significant difference (p>0.05) among the I/I, I/D, and D/D genotypes in blood pressure levels and other physiological characteristics needs a more comprehensive discussion. It would be insightful to further explore and discuss why these findings are significant and how they contribute to our understanding of ACE gene polymorphism in the population studied.

Author’s response: I agree with this suggestion and so following the advice, I have included additional discussion on this topic in the Discussion section as follows:

“.........Further, no significant association (p>0.05) between ACE gene polymorphism and physiological characteristics was observed among Monpa, which suggests that ACE gene doesn’t influence blood pressure, lung capacities and/or [Hb] concentration in the population studied...”(p. 12 & 13).

“......This phenomenon perhaps indicates that ACE gene doesn’t influence arterial blood pressure in the studied population....” (p. 13).

5. Language and Formatting: Finally, there are minor language and formatting issues that need to be addressed to enhance the readability and professionalism of the manuscript.

Author’s response: Following this suggestion, I have taken professional help to edit the manuscript. 

Additional Editor’s Comments:

Author’s response: I am thankful to the additional editor for the comments. I have incorporated all the suggestions and given point-by-point answers in the above section.

---

## [Editor Report · Decision Letter 1]

7 Sep 2023

Angiotensin Converting Enzyme (ACE) gene polymorphism and arterial blood pressure among the Tawang Monpa of Eastern Himalayan Mountains: Is there a signature of natural selection?

PONE-D-23-15795R1

Dear Dr. Ghosh,

We’re pleased to inform you that your manuscript has been judged scientifically suitable for publication and will be formally accepted for publication once it meets all outstanding technical requirements.

Kind regards,

Esteban Ortiz-Prado

Academic Editor

PLOS ONE

Additional Editor Comments (optional):

Dear Dr. Sudipta Ghosh,

I am pleased to inform you that after a thorough review and consideration of the revisions you made in response to the reviewers' comments and suggestions, I am inclined to accept your manuscript titled "Angiotensin Converting Enzyme (ACE) gene polymorphism and arterial blood pressure among the Tawang Monpa of Eastern Himalayan Mountains: Is there a signature of natural selection?" for publication in the PLOS Journal.

However, I'd like to highlight a few minor modifications that would further refine the quality of the manuscript:

Language and Clarity: There are sporadic instances where the phrasing of sentences could be enhanced for better readability. I recommend a final pass for sentence structure and grammar.

Table Presentation: Ensure all tables have appropriate titles and legends for clarity, and that they are consistently formatted.

Figure Labels: Some figures could benefit from clearer labeling, ensuring all abbreviations are defined and discernible to readers unfamiliar with the topic.

References: Double-check that all citations in the text are present in the references list and vice versa. Also, ensure a consistent formatting style throughout the references section.

Although these changes are minor, I believe they will enhance the overall presentation and clarity of the manuscript. Once these final edits are made, I am confident that the manuscript will be ready for publication.

Our editorial team will assist with the final preparation of your manuscript. Once your corrections are received and approved, you will be contacted by the production department with proofs for your review and more details on the forthcoming steps.

Thank you for your dedication to improving the manuscript based on the feedback provided. I trust that your research will be a notable contribution to the scientific community and a valuable resource for our readers.

Congratulations on your work and best wishes for your future research endeavors.

Warm regards,

Esteban Ortiz Prado, MD, PhD

Editor

PLOS Journal
---

## [Editor Report · Acceptance letter]

12 Sep 2023

PONE-D-23-15795R1 

Angiotensin-Converting Enzyme (ACE) gene polymorphism and arterial blood pressure among the Tawang Monpa of Eastern Himalayan Mountains: Is there a signature of natural selection? 

Dear Dr. Ghosh:

I'm pleased to inform you that your manuscript has been deemed suitable for publication in PLOS ONE. Congratulations! Your manuscript is now with our production department. 

Kind regards, 

on behalf of

Dr. Esteban Ortiz-Prado 

Academic Editor

PLOS ONE